# Designs for Enabling Collaboration in Human-Machine Teaming via Interactive and Explainable Systems

**Rohan Paleja**
MIT Lincoln Laboratory
Lexington, MA 02142
rohan.paleja@ll.mit.edu

**Michael Munje**
The University of Texas at Austin
Austin, TX 78712
michaelmunje@utexas.edu

**Kimberlee Chestnut Chang, Reed Jensen**
MIT Lincoln Laboratory
Lexington, MA 02142
{chestnut, rjensen}@ll.mit.edu

**Mathew Gombolay**
Georgia Institute of Technology
Atlanta, GA 30332
matthew.gombolay@cc.gatech.edu

## Abstract

Collaborative robots and machine learning-based virtual agents are increasingly entering the human workspace with the aim of increasing productivity and enhancing safety. Despite this, we show in a ubiquitous experimental domain, Overcooked-AI, that state-of-the-art techniques for human-machine teaming (HMT), which rely on imitation or reinforcement learning, are brittle and result in a machine agent that aims to decouple the machine and human's actions to act independently rather than in a synergistic fashion. To remedy this deficiency, we develop HMT approaches that enable iterative, mixed-initiative team development allowing end-users to interactively reprogram interpretable AI teammates. Our 50-subject study provides several findings that we summarize into guidelines. While all approaches underperform a simple collaborative heuristic (a critical, negative result for learning-based methods), we find that white-box approaches supported by interactive modification can lead to significant team development, outperforming white-box approaches alone, and that black-box approaches are easier to train and result in better HMT performance, highlighting a tradeoff between explainability and interactivity versus ease-of-training. Together, these findings present three important future research directions: 1) Improving the ability to generate collaborative agents with white-box models, 2) Better learning methods to facilitate collaboration rather than individualized coordination, and 3) Mixed-initiative interfaces that enable users, who may vary in ability, to improve collaboration.

## 1 Introduction

Successful human-machine teaming (HMT) has long been sought after for its wide utility across potential applications, ranging from virtual agents such as "clippy" that provide on-demand support

---

DISTRIBUTION STATEMENT A. Approved for public release. Distribution is unlimited. This material is based upon work supported by the Under Secretary of Defense for Research and Engineering under Air Force Contract No. FA8702-15-D-0001. Any opinions, findings, conclusions or recommendations expressed in this material are those of the author(s) and do not necessarily reflect the views of the Under Secretary of Defense for Research and Engineering. Delivered to the U.S. Government with Unlimited Rights, as defined in DFARS Part 252.227-7013 or 7014 (Feb 2014). Notwithstanding any copyright notice, U.S. Government rights in this work are defined by DFARS 252.227-7013 or DFARS 252.227-7014 as detailed above. Use of this work other than as specifically authorized by the U.S. Government may violate any copyrights that exist in this work.

for improving documents to embodied robotic healthcare aides that can provide doctors with a helping hand [29]. While promising, achieving fluent HMT is challenging because interactions with humans can be incredibly complex due to the diversity across users [26], human teammates benefit from explainable systems to support the development of mental models [31], and the lack of bidirectional communication (i.e., unclear how humans can "tell" a machine online to perform a desired behavior) [46]. In this paper, we transition from the conventional approach of crafting an HMT solution that aims for flawless out-of-the-box performance to a paradigm where end-users can actively interact with and program AI teammates, fostering a more dynamic and developmental interaction between humans and AI. Specifically, we explore enabling humans to perform user-specific modifications to a collaborative AI's interpretable policy representation across repeated iterations of teaming episodes and provide a set of design guidelines to support team development in HMT drawn from a large-scale user study.

Recently, data-driven techniques (e.g., imitation and reinforcement learning) have become popular in HMT, allowing for the generation of collaborative agent behavior without cumbersome manual programming [40, 5]. However, these prior works utilize opaque, black-box models, limiting human's ability to develop a shared mental model and maintain situational awareness [27], crucial for high-performance teaming [37]. We posit that successful, real-world HMT is not feasible without the use of white-box methods, especially in safety-critical domains such as healthcare and manufacturing. Furthermore, collaborative interactions with machines have often lacked the ability to effectively learn with and adapt to human teammates in real-time [19]. In ad hoc human-human teams, effective teaming is often developed through an iterative process [43]. Bi-directional communication is often a key component of this process, enabling the development of successful coordination strategies [36]. In our work, we build towards such a team development paradigm in HMT by 1) creating a pathway of bi-directional communication, utilizing interpretable policy representations as a mechanism to allow users to understand their machine teammates and allowing for explicit teammate policy modification through an interface (users can modify the machine's tree-based policy via a GUI), and 2) allowing for the process of iterative mixed-initiative team development through repeated teaming episodes. We believe this paradigm is necessary because human-partnered systems need explainable components and adaptable systems. We provide the following contributions:

- We provide a case study regarding prior work in HMT [5, 40], finding that the generated machine behavior is unable to adapt to human-preferred strategies, and that high performance is typically driven by independent machine actions rather than collaboration, which can ultimately result in a higher team score.
- We create a novel InterpretableML architecture to support the creation of tree-based cooperative agent policies via reinforcement learning and a GUI to allow users to modify the AI's behavior to their specifications. This capability is promising, enabling end-users to "go under-the-hood" of machine learning models and tune affordances or interactively and iteratively reprogram behavior.
- We conduct a 50-participant between-subjects user study assessing the effects of interpretability and interactive policy modification across repeated interactions with an AI. We summarize our study findings into a set of design guidelines to support future HMT research.

## 2    Preliminaries

Here, we introduce prior work in HMT and Explainable AI, our experimental domain, Overcooked-AI, a model of team development used to understand our findings, Tuckman's Model, and the mathematical framework under which we generate agents, Markov Games.

**Human-Machine Teaming –** The field of HMT is concerned with understanding, designing, and evaluating machines for use by or with humans [6, 44, 30]. A popular technique that has been used to produce collaborative AI agents is Reinforcement Learning (RL) [28], where researchers have concentrated efforts on reducing the dissimilarity between synthetic human training partners and testing with human end-users. Approaches that have achieved some success include utilizing human gameplay data to finetune simulated training partners to behave more human-like [5], which can be expensive, and training with a diverse-skilled population of synthetic partners to create an agent that can better generalize to non-expert end-users [40], which may bias the AI teammate to exhibit individualized strategies, as we display in Section 3. *We note our work focuses on an interaction different from AI-assisted decision-making or decision support. Here, a human and an agent must collaborate across a series of timesteps, aiming to maximize a multifaceted joint objective function.*

**Explainable AI –** xAI is concerned with understanding and interpreting the behavior of AI systems [23]. In our work, we follow recent trends that show black-box methods paired with local explanations can be harmful [34] and utilize interpretable, *white-box tree-based models* in a multi-agent sequential decision-making problem. These models have been shown to be beneficial in improving the user's ability to simulate a decision-making model [42] and providing users with increased situational awareness over a teammate's behavior in an HMT setting [31]. While tree-based models can provide users insight into the model, the complexity of the tree-based model limits its utility [24]. While we note this as a potential weakness of utilizing tree-based models, effective state representations can provide a tradeoff between granular control and tree depth. Accordingly, we design our trees to reason over a state-space with high-level binary features and multi-step macro-actions, expanded on below. Furthermore, in our work, we explore a paradigm where a user can directly modify and visualize a tree-based AI teammate the user is interacting with after a teaming episode. Prior work in explainable debugging [18] and robotics [33, 9] has explored similar paradigms, creating interactive systems that allow end-users to modify agent behavior to increase performance, but has not explored deploying tree-based models trained via RL in a collaborative HMT setting. We provide a working definition of what we mean by "interpretable" within the Appendix Section G.

**Overcooked-AI –** Overcooked-AI [5] is a testbed to evaluate human-AI interaction and has been used across HMT research concerned with collaboration [40], teammate identification [14], intention prediction [45], and behavior influence [16]. Here, two agents are tasked with creating and delivering as many soups as possible within a given time. Achieving a high score requires agents to navigate a kitchen and repeatedly complete a set of sequential high-level actions, including collecting ingredients, placing ingredients in pots, cooking ingredients into a soup, collecting a dish, getting the soup, and delivering it. Both players receive the same score increase upon delivering the soup. *We modify the original Overcooked-AI game to be a simultaneous-move game as opposed to the original formulation of allowing agents to perform actions asynchronously.* This modification prevents the collaborative score metric from being dominated by super-human AI speed, causing the overall score to be more reliant upon effective collaboration and strategy. We provide details about the state and action space below and complete details in the appendix.

*State-Space:* Policies reason over a semantically meaningful feature space as opposed to pixel space, detailing the objects each agent is holding, pot statuses, and counter objects. This state space allows for learning an interpretable tree-based policy that can be understood and manipulated by end-users.

*Action-Space:* Instead of using cardinal actions, we allow the AI to utilize macro-actions that can accomplish high-level objectives such as ingredient collection, ingredient placement, and soup serving. Macro-actions are planned using an A* planner, and we perform dynamic replanning at each timestep. Constructing trees on a higher level of abstraction results in smaller trees that are easier to interpret.

**Tuckman's Model –** Tuckman [43] describes the different stages that a team goes through before reaching high performance, including "Forming", "Storming", "Norming" and "Performing," often seeing a drop in performance as team members acclimate, followed by a rise as team members understand how to collaborate. Assuming that human-machine teams will follow similar stages to human-human teams, this paper looks into how we can support human-machine teams in reaching the Performing stage, where the team is achieving its full potential and exhibiting the highest level of cooperation. We provide a depiction of these stages as part of Figure 2.

**Markov Game –** We formulate our setting as a Markov Game [25], defined by a set of global states, $S_1, S_2 \in S$, a set of actions, $A_1, A_2 \in A$, transition function, $T : S \times A_1 \times A_2 \mapsto S$. and reward function $r_i : S \times A_i \mapsto \mathbb{R}$. Agent $i$ aims to maximize its discounted reward $R_i = \sum_{t=0}^{T} \gamma^t r_i^t$, where $\gamma \in [0, 1]$ is a discount factor. For training, we utilize agent-agent collaborative training, which trains two separate agents jointly via single-agent PPO. We utilize PantheonRL [38] for training our agents, incorporating our novel tree-based architecture (Section 4.1) into the codebase.

## 3   A Gap in Teaming Performance

In this section, we present two examples to display a gap in the quality of AIs in HMT. Specifically, we look at two recent approaches to produce collaborative AI agents [40, 5]. We argue and display that the AIs trained via these approaches are rigid and exhibit individualized behaviors, missing out on collaborative teaming strategies that can ultimately result in higher team scores. *We require AI agents that can effectively reach a consensus with humans on a teaming strategy that ultimately results in high performance. In cases where the human has a preferred strategy, the AI teammate should be able to support said strategy.*

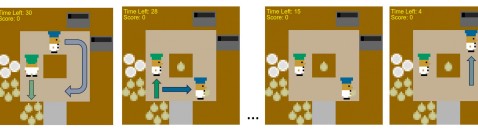
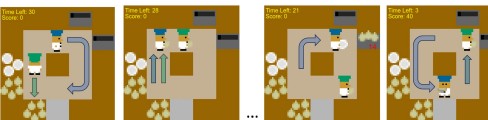
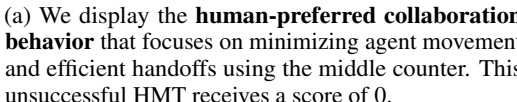

(a) We display the **human-preferred collaboration behavior** that focuses on minimizing agent movement and efficient handoffs using the middle counter. This unsuccessful HMT receives a score of 0.

(b) We display a human adapting to an **AI-preferred suboptimal teaming strategy**, where agents act individually. This individualized coordination results in minor success, achieving a low score of 40.

Figure 1: Case Study in Human-Machine Teaming with Different Teaming Strategies. It is clear that the models are not robust to multiple strategies of play and can result in agents performing nonsensical behavior (e.g., stuck in place).

In Figure 1, we display the *Coordination Ring* scenario. A simple collaboration strategy (which we term "human-preferred") in this domain is to utilize the counter to continuously pass objects, minimizing agent movement through efficient handoffs. To test a set of collaboration strategies, we utilize agents publicly available from Carroll et al. [5]. In Figure 1, we display a frame-by-frame of the human-preferred coordination strategy (Figure 1a) and AI-preferred coordination strategy (Figure 1b), which was a strategy where agents act individually to collect ingredients and place them in pots. The latter behavior was inferred through repeated play with the publicly-available AI. With the human-preferred strategy, the AI agent freezes for the majority of the game, creating an extremely frustrating and low-performing AI teammate. In this scenario, the human (green) picks up an ingredient and places it on the counter at the start of the game. The AI agent (blue), unfamiliar with this teaming strategy, freezes for approximately $80\%$ of the remaining episode before finally placing an onion in the pot. With the AI-preferred strategy, the human is able to successfully team with the AI, with each agent retrieving and placing ingredients while moving in a clockwise motion, but the strategy is not optimal or what the human prefers. As the AI produced by Carroll et al. [5] is created via RL teaming human-like AI teammates, the generated behavior may not be ideal for the current teammate, especially if the current teammate's preferred strategy was not present in the original training dataset used to create human-like AI training partners. This highlights a need for systems that can *explain strategies* exhibited by trained agent policies and allow humans to adapt these pre-trained policies toward human-preferred behavior.

In a second example, we utilize the *Optional Collaboration* domain, displayed on the right-hand side of Figure 4b, which is also utilized in our human-subjects experiment. This domain was designed to incentivize collaboration, where creating mixed-ingredient dishes facilitated by agents passing ingredients across the central counter will result in a higher score per dish. Here, we program two intelligent deterministic heuristics: In the first, each agent acts completely individually, cooking single-ingredient dishes and serving. In the second, agents share ingredients, which costs additional timesteps, but are able to successfully cook mixed ingredient dishes. We find that the collaboration strategy achieves a $408$ cumulative team score, approximately $30\%$ more score compared to the individualized strategy of $306$. However, we find that trained policies under Ficticious Co-Play [40] exhibit similar team score to that of the individual coordination strategy and further, find that real human end-users collaborating with these agents are unable to far surpass the individual strategy score. As Strouse et al. [40] trains an agent to work well with a population of agents, where approximately a third of the diverse-skilled population of agents used in training are completely random agents, we posit that the teammate agent must compensate and exhibit individualized behavior, limiting the algorithm's ability to effectively learn effective team coordination strategies. In line with the first case study, the trained collaborative agent policies miss out on high-performance teaming behaviors, and thus, we need systems where humans can iteratively improve agent behavior online.

*Thus, in the rest of the paper, we look to explore xAI techniques as a mechanism for closing this gap and allowing agents within a human-machine team to facilitate collaborative strategies that outperform the individualized and rigid behaviors trained agents assume.*

## 4 Methodology

In this section, we first present our architecture for training interpretable AI teammates. We then present a contextual pruning algorithm, allowing for ease-of-training and enhanced interpretability for neural tree-based models. We display an overview of our training procedure as part of Figure 2.

## 4.1 Interpretable Discrete Control Trees

We create an interpretable machine learning architecture, Interpretable Discrete Control Trees (ID-CTs), that can be used directly with RL to produce interpretable teammate policies. Below, we briefly detail our architecture, as well as advancements to enhance ease-of-training and interpretability.

**Architecture** Our IDCTs are based on differentiable decision trees (DDTs) [41] – a neural network architecture that takes the topology of a decision tree (DT). DDTs contain decision nodes and leaf nodes; however, each decision node within the DDT utilizes a sigmoid activation function (i.e., a "soft" decision) instead of a Boolean decision (i.e., a "hard" decision). Each decision node, $i$, is represented by a sigmoid function, displayed as $y_i = (1 + \exp(-\alpha(\vec{w}_i^T \vec{x} - b_i)))^{-1}$. As this representation is difficult to interpret, Paleja et al. [32] presented differentiable crispification, which recasts each decision node to split upon a single dimension of the input feature and translates the outcome of a decision node so that the outcome is a Boolean decision rather than a set of probabilities. This, in turn, allows for an interpretable forward propagation through the model that traces down a single branch of a tree as well as gradient flow afforded by the straight-through trick to update parameters of the neural tree model. We utilize this approach to learn interpretable tree-based teammate policies via reinforcement learning.

We initialize our IDCTs to be symmetric DTs with $N_l$ decision leaves and $N_l - 1$ decision nodes. Each decision leaf is represented by a sparse categorical probability distribution over actions. At each timestep, a state variable is propagated through each decision node, split on a single decision rule, with the output being a Boolean causing the decision to proceed via the left or right branch until arrival at a leaf node. At each leaf node, we sample from the respective distribution to produce a macro-action (e.g., in Overcooked-AI, "get an onion" or "place ingredient on counter"). Further, we improve model predictability by applying an L1 norm loss over leaf node distributions to ensure sparsity, penalizing high entropy action distributions at a leaf[1]. *Importantly, the resultant representation after training is that of a simple decision tree with categorical probability distributions at each leaf node.*

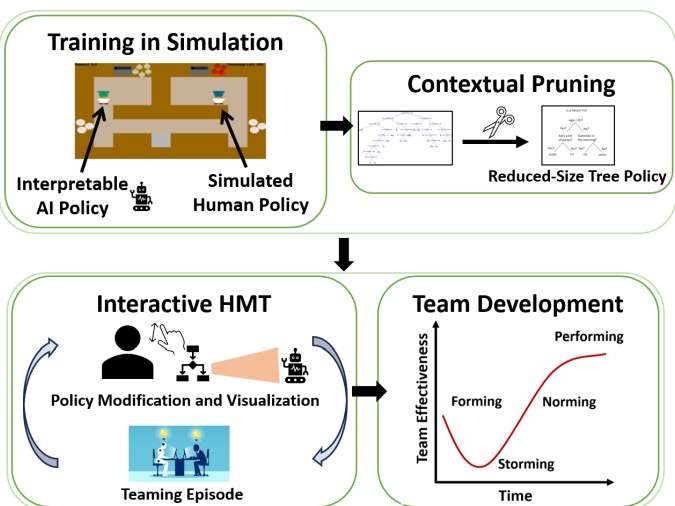

Figure 2: Here, we provide an overview of the steps to produce a collaborative AI teammate with an interpretable policy and the proposed policy modification scheme evaluated in our user study.

**Contextual Pruning** As we focus on creating agents that must cooperate with and be interpreted by humans, we must limit the size of our tree-based models to a certain depth to promote user understanding. Analogous to the "lottery ticket hypothesis" in network training that supports the practicality of employing large models [11], a small tree with a limited number of sub-trees (lottery tickets) may not have the representational power to learn a high-performing policy. Thus, the ability to effectively train IDCTs is at odds with maintaining user readability and simulatability. Following work in neural network pruning [22], we design a post-hoc *contextual pruning* algorithm that allows us to simplify large IDCT models while precisely adhering to model behavior by accounting for:

1. **Boundaries of a variable's state distribution**: We utilize the minimum and maximum of each variable's range to parse impossible subspaces of a tree.
2. **Node hierarchy**: Ancestor nodes for a specific decision node may have already captured a specific splitting criterion and, thus, may lead to redundancy. By detecting redundancies, we can prune subspaces of the tree.

---

[1]While utilizing deterministic AI policies may be easier to understand for users, we found these models could not converge to similar performance as the stochastic-leaf IDCT policies during training.

We provide further details and an algorithm for contextual pruning in the supplementary material. *This, in turn, allows us the benefit of training large tree-based models, greatly improving ease-of-training, while still being able to simplify the resultant model to a smaller, equivalent representation.*

## 4.2 Modifying an Interpretable Policy

While the above architecture can be used alongside RL to produce a collaborative AI policy, the result may not actually be helpful or what the human wants. *Humans, when teaming with machines, should be able to intuitively update what the robot has learned or change it based upon preferences that may evolve over time.* Such is critical in the positive development of coordination strategies and is associated with the calibration of trust, assignment of roles, and development of a shared mental model. As such, we propose a *policy modification scheme* that allows the user to repeatedly team with an AI maintaining an IDCT policy, visualize the current behavior in tree form, and modify its AI's behavior.

The iterative process generated through this scheme can facilitate a feedback loop, allowing for the possibility of team development and improved HMT performance over teaming episodes.

We term our modification scheme *human-led policy modification.* We provide humans with an explicit pathway to "communicate" with an AI after each teaming interaction through a GUI, with capabilities displayed in Figure 3. Within this interface, users start with the pre-trained collaborative AI IDCT policy and can modify the AI's behavior by creating a new tree structure that may vary in what state features appear in the decision nodes, actions taken in leaf nodes, and the respective probabilities of actions within the leaf node. It is important to note that users are limited to expanding the tree to a depth of four (i.e., a max of 16 leaves), and the modification is not timed.

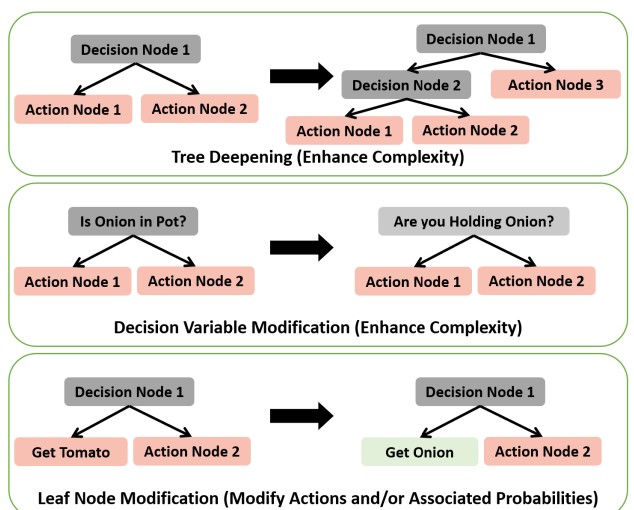

Figure 3: Users have several capabilities in creating an effective teammate, including modifying the tree structure by adding or removing decision nodes, changing state features the tree is conditioned on, and modifying actions and/or their respective probabilities at leaf nodes.

## 4.3 Trained Collaborative Teammate Policies

Across our experiment, we study collaboration in two domains, Forced Coordination and Optional Collaboration, displayed on the left-hand side of Figure 4. In each domain, we train an IDCT policy via agent-agent collaborative training and a neural network (NN) policy following the population-based training scheme in Strouse et al. [40]. In the first domain of Forced Coordination, the IDCT policy converged to a policy with an average reward of $315.22 \pm 14.59$, and the neural network policy converged to an average reward of $403.16 \pm 16.08$ evaluated over 50 teaming simulations with the synthetic human teammate the policy was trained with. In the second domain, Optional Collaboration, the IDCT policy converged to a policy with an average reward of $171.46 \pm 18.89$, and the neural network policy converged to an average reward of $295.02 \pm 1.86$. Thus, a consequent confound due to the current difference in performance capabilities between interpretable vs. black-box models is that the NN policy outperforms the IDCT policy in both domains. This displays a need for improving optimization algorithms for interpretable models representing collaborative agent policies. *However importantly, while the initial simulated performance of interpretable models may underperform black-box models, the ability for humans to understand machine behavior and improve upon behavior may allow these approaches to compete or even outperform black-box NN models.* We can also compare to the heuristic policies presented in Section 3, observing that the training performance of the IDCT and NN policies in the Optional Collaboration domain underperform the collaborative heuristic (408 vs. 295.02 and 171.46). We provide visualizations of the trained IDCT policies for

each domain in the appendix, finding that after contextual pruning, the AI IDCT policy has two and three leaves, respectively.

# 5 Human-Subjects Study

Here, we discuss our between-subjects user study that seeks to understand how users interact with an AI across repeated play under different factors. Below, we introduce our research questions, provide a description of the independent variables and procedure, and discuss our findings.

**Research Questions** The presented research questions below seek to understand changes in overall human-machine teaming performance and performance changes across repeated gameplay. The latter question pivots from an episodic attitude of teaming to a longer-term gauge, allowing us to study the process of adaptation in HMT.

1. **RQ1**: How does human-machine teaming performance vary across factors?
2. **RQ2**: How does team development vary across factors?

**Independent Variables** We have two independent variables, **IV1**: the teaming method, and **IV2:** the domain. For **IV1**, we consider the following conditions (abbreviated by **IV1-C**):

1. **IV1-C1: Human-Led Policy Modification:** After interacting with the agent (one teaming episode), the user can modify the policy via the GUI, allowing the user to update decision nodes and action nodes in the tree as well as tune affordances. Upon completion, the user can visualize the updated policy in its tree form prior to the next interaction.
2. **IV1-C2: AI-Led Policy Modification:** After interacting with the agent, the AI utilizes recent gameplay to fine-tune a human gameplay model via Behavioral Cloning and performs reinforcement learning for five minutes[2] to optimize its own policy to better support the human teammate. Upon completion of policy optimization, the user can visualize the updated AI policy in its interpretable tree form prior to the next interaction. This is similar to HA-PPO [5], adapted to an online setting.
3. **IV1-C3: Static Policy - Interpretability:** After interacting with the agent, the user can visualize the AI's policy in its interpretable tree form prior to the next interaction. *Throughout this condition, the AI's policy is static.*
4. **IV1-C4: Static Policy - Black-Box:** After interacting with the agent, the user does *not* see the AI's policy. *Here, the AI policy is the same as **IV1-C3**, but the human has lost access to direct insight into the model.*
5. **IV1-C5: Static Policy - Fictitious Co-Play:** [40]: User teams with an AI maintaining a static black-box, neural network (NN) policy trained across a diverse partner set. As this is a baseline, we utilize an NN rather than the legible IDCT policy used in other conditions (**IV1:C1-4**).

For **IV2**, we consider the following domains displayed on the left-hand side of Figure 4:

1. **IV2-D1: Forced Coordination:** Users team with an AI that is separated by a barrier and must pass over items in a timely manner. Here, agents are forced to collaborate.

2. **IV2-D2: Optional Collaboration:** In this domain, the team can operate individually or collaboratively. This domain has increased complexity, both with respect to the size of the domain and the types of soups that can be cooked. *Collaboration is incentivized through a higher reward for mixed-ingredient dishes (combining onions and tomatoes) over single-ingredient dishes.*

**Procedure:** A participant is first randomly placed into one of the five conditions in **IV1**. The participant starts with a pre-experiment survey collecting demographic information, experience with video games and decision trees, and the Big Five Personality Questionnaire [7]. Afterward, a participant conducts a brief tutorial in Overcooked with a random AI agent, improving

Table 1: A comparison across different **IV1** factors.

| Approaches | Explicit Interaction | Policy Changes Across Iterations | White-Box | Base Policy |
|---|---|---|---|---|
| **IV1-C1** | ✓ | ✓ | ✓ | IDCT |
| **IV1-C2** | ✗ | ✓ | ✓ | IDCT |
| **IV1-C3** | ✗ | ✗ | ✓ | IDCT |
| **IV1-C4** | ✗ | ✗ | ✗ | IDCT |
| **IV1-C5** | ✗ | ✗ | ✗ | NN |

---

[2]We limit the online optimization time for the AI teammate to five minutes to create a feasible user-study. This RL optimization is challenging as only a limited number of samples can be obtained in this time, and thus, the policy is not guaranteed to improve. In cases where the policy degrades, we use the original policy prior to optimization.

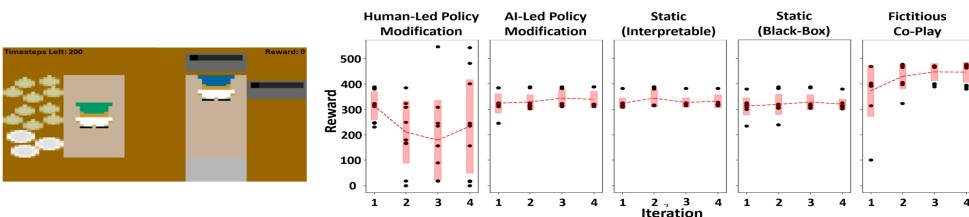

(a) Performance Data in **IV2-D1**: Forced Coordination.

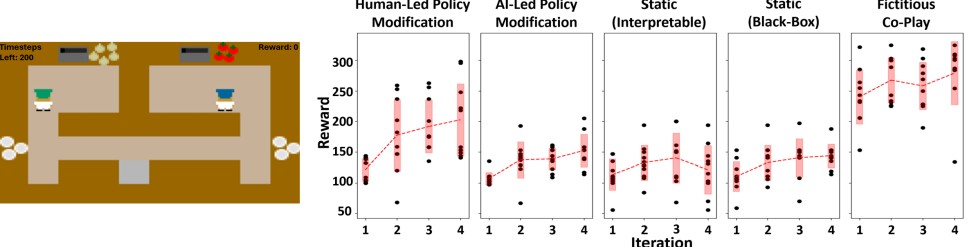

(b) Performance Data in **IV2-D2**: Optional Collaboration

Figure 4: User gameplay scores across teaming iterations with per-iteration means connected by the red dotted line and the per-iteration standard deviation shaded in red.

the user's understanding of game controls and the assigned task. Once completed, the primary experimentation begins. Users will team with an AI four times in each domain (randomly ordered), starting with the unique domain-specific pre-trained agent, and are told that their goal is to maximize their score in the last teaming interaction, the "performance round." After each teaming interaction, in the first three factors, the user will modify and visualize the AI's policy (**IV1-C1**), the AI will optimize its own policy proceeded by user visualization (**IV1-C2**), or the user will solely view the policy (**IV1-C3**). In **IV1-C4** and **IV1-C5**, as the AI is black-box (perceived to be black-box in **IV1-C4** and truly black-box in **IV1-C5**), transitionary pages are shown to the participant, providing them a pause before they team with the agent again. Upon completion of the condition-specific (or lack of) actions, users complete a NASA-TLX Workload Survey. After users have completed a domain, providing us with four episodes of teaming data and workload assessments, we administer several post-study scales, including the Human-Robot Collaborative Fluency Assessment [15], Inclusion of Other in the Self scale [1], and Godspeed Questionnaire [2]. Upon completion of the two domains, the experiment concludes.

## 5.1 Results

Our experiment is a 5 (teaming method; between-subjects) $\times$ 2 (no. of domains; within-subjects) $\times$ 4 (no. of repeated evaluations; within-subjects) mixed-factorial experiment. We recruited 50 participants under an IRB-approved protocol, whose ages range from 18 to 32 (Mean age: 24.14; Std. Dev.: 4.10; 46% Female, 52% Male, 2% Non-Binary), with participants randomly assigned to each of the factor levels, with ten total subjects per level. The duration of the experiment was $70.98 \pm 19.71$ minutes [3]. Our data was modeled as a full-factorial, between-subjects ANOVA. We test for normality and homoschedascity (see appendix) and employed a corresponding non-parametric test if the data failed to meet these assumptions. We display our objective findings in the right-hand side of Figure 4.

**RQ1: Team Performance:** In analyzing reward, we find trends with respect to the maximum reward participants obtained within a domain across iterations (Figure 5). Using Friedman's test, we find a significant difference across domains ($\chi^2(1)$=46.08, $p < 0.001$) and analyze the domains separately.

---

[3]The significant variance in experiment duration arises from the granularity across our conditions. The increase in human effort to understand and interact with the policy results in an increase in duration. We note that as our experiment is relatively short, it is unlikely that experiment fatigue played a role in our results as would be common in experiments with large task variances.

In **IV2-D1**, a Kruskal-Wallis Test was conducted to analyze differences in maximum performance obtained across teaming paradigms, finding a significant effect ($\chi^2(4) = 20.146, p < 0.001$). We conduct post-hoc pairwise comparisons, utilizing Dunn's test, and find that **IV1-C5** (Fictitious Co-Play) is significantly better than **IV1-C1** ($p < 0.001$), **IV1-C2** ($p < 0.01$), **IV1-C3** ($p < 0.01$), and **IV1-C4** ($p < 0.05$). Even though Fictitious Co-Play (**IV1-C5**) outperformed the tree-based models, likely due to its ability to converge to a higher-performance teaming policy, it is interesting that Human-Led Policy Modification (**IV1-C1**) has several participants that outperform the maximum performance of **IV1-C5** in teaming iterations three and four (Figure 4a).

In **IV2-D2**, a Kruskal-Wallis Test was conducted to analyze differences in participant teaming performance across conditions, finding a significant effect ($\chi^2(4) = 29.922, p < 0.001$). We conduct post-hoc pairwise comparisons, utilizing Dunn's test, and find that **IV1-C5** (Ficticious Co-Play) is significantly better than **IV1-C2** ($p < 0.001$), **IV1-C3** ($p < 0.001$), and **IV1-C4** ($p < 0.001$), and **IV1-C1** (Human-Led Policy Modication) is significantly better than **IV1-C2** ($p < 0.05$), **IV1-C3** ($p < 0.05$), and **IV1-C4** ($p < 0.05$). For white-box AI teammates (**IV1:C1-3**), the latter finding displays the benefit of Human-Led Policy Modification in improving HMT performance for interpretable models. These findings display that 1) white-box approaches supported with policy modification can outperform white-box approaches alone, 2) black-box models can outperform white-box approaches in HMT, and 3) by comparing **IV1-C3** to **IV1-C4**, interpretability alone afforded via tree visualizations did not provide any direct objective benefits. Finally, in Optional Collaboration, across all conditions we see that HMT scores are not near that of the collaborative heuristic, displaying a gap that must be addressed to achieve effective HMT.

**RQ2: Team Development:** In analyzing RQ2, we look at the change in reward across iterations one to four and relate our findings to Tuckman's model. Utilizing a Friedman's test, we find a difference across domains ($\chi^2(1)$=20.48, p<0.001) and analyze the domains separately. In **IV2-D1**, we see that none of the conditions results in a significant improvement in teaming performance over repeated iterations. In **IV2-D2**, we see **IV1-C1** ($p < 0.01$) and **IV1-C2** ($p < 0.01$) significantly improve over repeated teaming interactions. The improving interactions can be connected to the Norming stage in team development, where teams begin to develop a strategy and team mental models. *We see conditions that facilitate Norming have the attribute of policy adaptation and are white-box.*

Next, we analyze whether different person-specific factors allow HMT to improve more quickly than others. In **IV2-D1**, we find that conscientiousness is trending in its correlation with improvement ($0.05 < p < 0.1$). In **IV2-D2**, we find that participants with high familiarity with Trees improve more across iterations ($F(1) = 7.448, p < 0.01$). These findings signify that positive interaction with interpretable models may be more beneficial to those with an engineering background and specific personality traits.

Finally, we detect an interesting trend in **IV2-D1** under the **IV1-C1** condition. We see a drop in performance between the first teaming iteration and later iterations, followed by a rise. We believe this relates to the Forming and Storming stages, where team members are still develop-

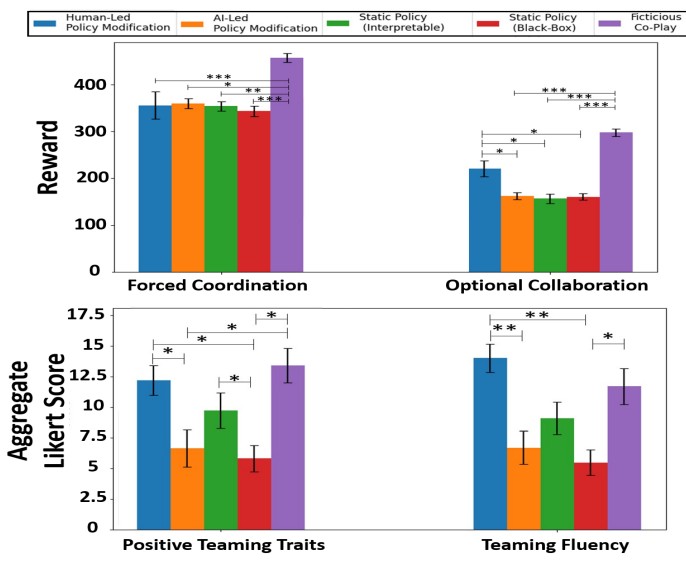

Figure 5: Maximum Reward and Subjective Ratings Across **IV1** Factors.

ing effective strategies to coordinate. As the last iteration shows an improvement in performance, we hypothesize that the team was shifting into the Norming stage. In future, it would be interesting to evaluate a larger number of iterations to see if the behavior would continue to uptrend. This requires further research due to the additional resources and time needed for more teaming iterations.

**Subjective Findings:** In **IV2-D1**, we find that users did not find any subjective differences toward the teaming interaction across conditions. In **IV2-D2** (Figure 5), we find that users find collaboration with AIs under condition **IV1-C2** and **IV1-C4**, on average as less fluent than **IV1-C1** (p<0.01, p<0.01), and **IV1-C4** as less fluent than **IV1-C5** ($p < 0.05$). Users also trusted the AI and perceived the AI contributed more in **IV1-C5** than in **IV1-C2** (p<0.05, p<0.05) and **IV1-C4** ($p < 0.05, p < 0.05$). Furthermore, the users viewed the AI more positively in **IV1-C1** and **IV1-C5** than in both **IV1-C2** (p<0.05, p<0.05) and **IV1-C4** (p<0.05, p<0.01). Overall, participants generally assessed higher-performing agents more positively in their subjective ratings. In considering conditions that utilized a tree-based model (**IV1-C1, IV1-C2, IV1-C3, and IV1-C4**), we see the addition of interaction with the tree policy provides significant subjective benefits in positive teaming traits and collaborative fluency (defined within the Human-Robot Collaborative Fluency Assessment [2]). In including the remaining condition, which utilizes a black-box model, **IV1-C5: Fictitious Co-Play**, and comparing it to **IV1-C1: Human-Led Policy Modification**, we see that even though Fictitious Co-Play outperformed Human-Led Policy Modification in terms of team reward (though not significantly in the domain of Optional Collaboration), no significant subjective differences were observed between these two conditions. This presents an interesting relationship between transparency, interaction, and performance in relation to subjective perception that warrants future research.

**Design Guidelines:** To achieve fluent HMT, we specify the following forward-facing guidelines.

1. *The creation of white-box learning approaches that can produce interpretable collaborative agents that achieve competitive initial performance to that of black-box agents.* This guideline is critical to providing humans with the subjective benefits obtained from interactivity with white-box models, objective benefits of black-box models, and the ability to interact with policies to facilitate team development.
2. *The design of learning schemes to support the generation of collaborative AI behaviors rather than individual coordination.* We need techniques that avoid converging to the local maxima of individual coordination and scenarios that allow for properly evaluating cooperation.
3. *The creation of mixed-initiative interfaces that enable users, who may vary in ability and experience, to improve team collaboration across and within interactions.* As we found a large diversity in perceived usability of our interface (finding an average score of $58.25 \pm 27.61$, with some users finding the interface good (>75) and others poor (<35)), effective interfaces are vital in shifting from only a subset of users benefiting to all users being able to create effective teammates.
4. *The evaluation of teaming in a larger number of interactions.* As agents are deployed, team performance will change over time, going through a transient period before reaching peak performance. Understanding this process of team development is essential in creating high-performance HMT.

## 6 Conclusion

This work investigates repeated interactions with machine learning models within a sequential decision-making HMT paradigm. We present a key gap in HMT, displaying that current methods do not facilitate human-machine collaboration to the fullest. We find that human-led policy modification allows for a team to achieve higher performance than white-box models without this capability. However, as interpretable models are more difficult to generate, Fictitious Co-Play is able to better support high performance. Given these mixed findings, future work must focus on developing better white-box teammates, study the modality of communication in HMT, and explore mechanisms to allow HMT to scale beyond individual coordination and toward effective collaboration.

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

# A  Appendix

In the Appendix, we provide further information regarding our testbed for Human-Machine Collaboration, Overcooked-AI (Section B), additional model and training details for our interpretableML architecture, the Interpretable Discrete Control Tree (Section C), complete information regarding our statistical analysis (Section E), further discussion regarding our paper's results, limitations, and future work (Section F), and finally, a working definition of what we mean by "interpretable".

# B  Overcooked-AI

Overcooked-AI [5] is a testbed to evaluate human-AI interaction and has been used across numerous prior work studying human-AI collaboration [40, 10]. Here, two agents are tasked with creating and delivering as many soups as possible within a given time. Achieving a high score requires agents to navigate a kitchen and repeatedly complete a set of sequential high-level actions, including collecting ingredients, placing ingredients in pots, cooking ingredients into a soup, collecting a dish, getting the soup, and delivering it. Both players receive the same score increase upon delivering the soup. *We modify the original Overcooked-AI game to be a simultaneous-move game as opposed to the original formulation of allowing agents to perform actions asynchronously.* This modification prevents the collaborative score metric from being dominated by super-human AI speed, causing the overall score to be more reliant upon effective collaboration and strategy.

We utilize two map configurations we term, *Forced Coordination* and *Optional Collaboration*, displayed in Figure 4 of the main paper. Each domain was chosen so that collaborating with the teammate would result in a higher score than working individually. In our newly-created domain, Optional Collaboration, creating mixed-ingredient dishes (combining onions and tomatoes) will receive a higher score than single-ingredient dishes. Teammates have 200 timesteps to collaborate and cook as many dishes as possible.

**State-Space, Action-Space, and Reward Scheme**    Policies reason over a semantically meaningful 13-dimensional feature space as opposed to pixel space, detailing the objects each agent is holding, pot statuses, and counter objects. Each of these features is binary. For the action space, instead of using cardinal actions, we allow the AI to utilize macro-actions that can accomplish high-level objectives such as ingredient collection, ingredient placement, and soup serving. Macro-actions are planned using an A* planner, and we perform dynamic replanning at each timestep. Prior work has shown macro-actions can enhance interpretability [4]. This state and action space allow forlearning an interpretable tree-based policy that can be understood and manipulated by end-users.

In Forced Coordination, for the reward scheme, we follow a similar distribution as prior work and give a reward score of 60 per dish served, 3 for an item placed into a pot, 3 for a useful dish pickup, and 5 for a soup pickup. In Optional Collaboration, for the reward scheme, we give a reward score of 50 for a mixed-ingredient dish, 30 for a single ingredient dish, 3 for an item placed into a pot, 3 for a useful dish pickup, and 5 for a soup pickup.

# C    Additional IDCT Model Details

Here, we provide additional model details for the proposed Interpretable Discrete Control Tree (IDCT).

## C.1    Architecture

Our IDCTs are based on differentiable decision trees (DDTs) [41] – a neural network architecture that takes the topology of a decision tree (DT). DDTs contain decision nodes and leaf nodes; however, each decision node within the DDT utilizes a sigmoid activation function (i.e., a "soft" decision) instead of a Boolean decision (i.e., a "hard" decision). Each decision node, $i$, is represented by a sigmoid function, displayed as $y_i = \frac{1}{1+\exp(-\alpha(\vec{w}_i^T \vec{x} - b_i))}$, where $\vec{w}_i$ and $b_i$ represents the weight and bias terms of the decision node, respectively. As this representation is difficult to interpret, [32] presented differentiable crispification, consisting of two components: 1) Decision node crispification, which recasts each decision node to split upon a single dimension of our input feature, and 2) Decision outcome crispification, which translates the outcome of a decision node so that the outcome is a Boolean decision rather than a set of probabilities. Both operations utilize the straight-through trick [3] to maintain gradients, allowing for both an interpretable forward propagation through the model that traces down a single branch of a tree as well as gradient flow to update parameters of the neural tree model. We utilize this approach in our IDCTs to maintain interpretability.

We initialize our IDCTs to be symmetric complete decision trees with $N_l$ decision leaves and $N_l - 1$ decision nodes. Each decision leaf is represented by a sparse categorical probability distribution over actions. At each timestep, a state variable is propagated through each decision node, split on a single decision rule, with the output being a Boolean causing the decision to proceed via the left or right branch until arrival at a leaf node. At each leaf node, we sample from the respective probability distribution to produce a macro-action (e.g., "get an onion" or "place held ingredient on counter").

## C.2    Training

For training this model, we utilize agent-agent collaborative training where an interpretable tree-based agent (maintaining an IDCT) is paired with a second policy (representing the human player), and both models are trained via decentralized PPO [39]. It is important to note that each agent maintains its own buffer and optimizers. Further, we improve model predictability by applying an L1 norm loss over leaf node distributions for the IDCT agent to ensure sparsity, penalizing high entropy action distributions at a leaf. Our training procedure mimics that of PPO, utilizing a modified loss function displayed in Equation 1, and policy update in Equation 2, where $\theta$ represents the aggregate set of weights for the IDCT, $\hat{A}_t$ represents the advantage estimate at time $t$, and $a_l$ represents the distribution maintained at each leaf, $l$.

$$L(\theta) = \mathbb{E}_\tau \left[ \min \left( r_t(\theta) \hat{A}_t, \mathrm{clip}\left( r_t(\theta), 1 - \epsilon, 1 + \epsilon \right) \hat{A}_t \right) \right]$$
$$+ \sum_1^L \lambda |a_l| \tag{1}$$

$$\theta_{k+1} = \arg\max_\theta L(\theta) \tag{2}$$

### C.3 Contextual Pruning

As we focus on creating agents that cooperate with humans, we must limit the size of our interpretable tree-based models to a certain depth to promote user understanding. This follows prior work, finding trees of arbitrarily large depths can be difficult to understand [13] and simulate [24], and that a sufficiently sparse DT is desirable and considered interpretable [20]. However, this can make training difficult, as a small tree may not have the representational power to learn a high-performing policy.

---

**Algorithm 1** Contextual Pruning Algorithm

---

**Input**: IDCT I(.)
**Output**: Pruned IDCT

1: SET_NODE_DOMAINS(IDCT=I, minValue=0, maxValue=1)
2: queue = [I.root]
3: **while** queue is not empty **do**
4:     currentNode ← queue.pop()
5:     **if** currentNode.compareValue < currentNode.lowerBound **then**
6:         currentNode.prunable = True
7:     **end if**
8:     **if** currentNode.compareValue > currentNode.upperBound **then**
9:         currentNode.prunable = True
10:     **end if**
11:     UPDATE_DOMAINS_FOR_CHILDREN(currentNode,     lowerBound,     upperBound,     currentNode.compareValue)
12:     ADD_CHILDREN_TO_QUEUE(currentNode, queue)
13: **end while**
14: I ← PRUNE_NODES_FROM_TREE(I)
15: **return** I

---

In Algorithm 1, we present details of how contextual pruning is accomplished. In Step 1, we initialize a domain vector representing the current minimum and maximum values for each feature. Since our Overcooked domain utilizes binary features, all bounds are initialized to 0 and 1. Formally, this can be written as by the Cartesian product $B = [0, 1] \times \cdots [0, 1]$, of cardinality $d$ (where $d$ is the dimensionality of the state space). In Step 2, we initialize a queue that will be used to perform a breadth-first search to visit each node in a hierarchical order. In Step 4, we receive a node from the queue. In Step 5, we check the threshold value of the current node and compare it to the current node's vector of minimum values. This operation looks to see if the node results in a tree sub-space that is out of bounds (i.e., impossible to reach). We perform a similar computation in step 8, checking the maximum values. In both cases, we look to find child nodes that do not yield a reduction in the hyperspace as candidates for pruning. In Step 11, we update the children based on the threshold value of our current node and its sign (as we can have < or > within a node), creating a new bounding box. In step 12, we add the children of the current node to the queue, and loop back to Step 4, repeating steps 5-12 until the queue is empty. In Step 14, we prune tree sub-spaces that are impossible to reach.

### C.3.1 Computational Analysis

The computational complexity of our contextual pruning algorithm can be analyzed in terms of both time and space complexity. In terms of time complexity, it is equivalent to that of Breadth-First Search (BFS), specifically, $\mathcal{O}(V + E)$, where V denotes the number of vertices and E represents the number of edges in the tree. Regarding space complexity, our algorithm exhibits similar characteristics to BFS for trees with only two leaves. In such cases, the space complexity of BFS is $\mathcal{O}(V)$, as it stores

all the vertices at the maximum breadth level in the queue during the traversal. Consequently, the space complexity of our contextual pruning algorithm is also $\mathcal{O}(V)$, making it efficient and scalable for trees with a limited number of leaves.

*Utilizing contextual pruning alongside our training framework allows us the benefit of training large tree-based models, greatly improving ease-of-training, while still being able to simplify the resultant model to a smaller, equivalent representation.*

### C.3.2 Results of Pruning

To evaluate the utility of pruning, we train models of various sizes (8-leaf, 16-leaf, 32-leaf, 64-leaf, 128-leaf, 256-leaf) in Forced Coordination and perform pruning on the resultant model. We find that models of larger size converge to higher performance (i.e., easier-to-train), following prior work displaying the utility of larger models. Further, empirically, we find we can reduce model sizes by 64-128x in tree depth. We provide a pipeline to allow for model training and contextual pruning in our GitHub repository https://github.com/CORE-Robotics-Lab/Team-Development-with-Transparent-Policies.

### C.4 Hyperparameters

In **IV2-D1**: Forced Coordination and **IV2-D2**, we train an IDCT with 256 leaves, a learning rate of $1e^{-3}$, and regularization parameter of $1e^{-4}$. This hyperparameters were chosen through trial and error, where we find larger models with a small learning rate and regularization exhibited greater learning early on. The rest of the parameters follow default parameters from the PantheonRL codebase [38] for training Overcooked agents. After contextual pruning, in both domains, we end up with an AI policy with two and three leaves in Forced Coordination and Optional Collaboration, respectively.

For training fictitious co-play agents, we train 32 models of teammates in each domain, saving policies at every 100 epochs. At the end of training, we sort the performance of saved policies and utilize the initial, mid-performing, and highest to create our population of diverse agents, totaling 96 agents. A neural network model is then paired in a multi-task training framework to team with this agent.

Our models are all trained on a local desktop computer containing a Nvidia RTX 2080 GPU and 16 GB of CPU memory. Training time for each agent took approximately 12 hours across a single core. We provide further instructions to replicate our models within the above codebase.

We include a high-level diagram of how IDCT agents are generated in Figure 6.

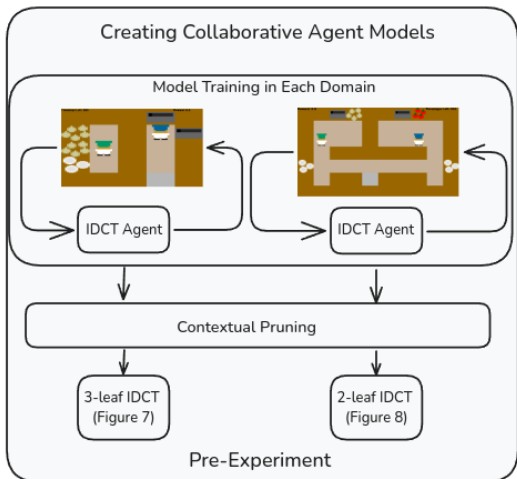

Figure 6: Tree Policy Generation for Conditions IV1-C1-C4

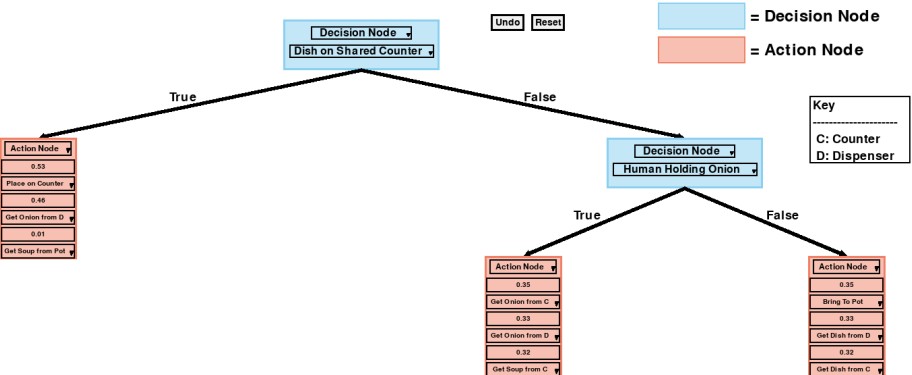

Figure 7: Trained Interpretable Discrete Control Tree in the Forced Coordination Domain.

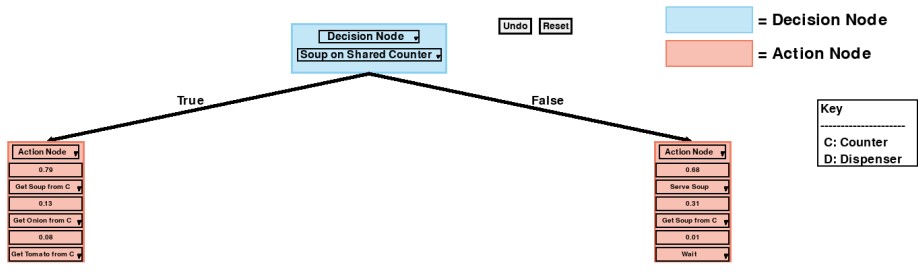

Figure 8: Trained Interpretable Discrete Control Tree in the Optional Collaboration Domain.

## C.5 Visualization of IDCT Policies in Each Domain

Here, we present visualizations of trained IDCT models in each domain. As seen in Figures 7 and 8, the resultant policies have two and three leaves for the Forced Coordination and Optional Collaboration domains, respectively. Note that these images are pulled from our interface and thus have extra annotations to improve readability.

# D   Additional User Study Information

Our experiment was reviewed and approved by the Institutional Review Board at the Georgia Institute of Technology under Protocol Number H23043. All participants in our experiment signed a consent form, received a description of the risks involved in our study, and received compensation for participating. Below, we describe specifics regarding the consent procedure, additional details that describe the experiment procedure, and the compensation scheme.

## D.1   Consent Procedure

At the start of the experiment, the participant is provided a consent document. This document describes the purpose of the experiment, exclusion/inclusion criteria, the experiment procedure, the risks of the experiment, the compensation scheme, and details regarding data storage and confidentiality.

## D.2   Additional Information Regarding Specific Conditions

**IV1-C1: Human-Led Policy Modification** is enabled through the contribution of the interpretable machine learning architecture to train collaborative AI teammates, a training advancement to enhance

interpretability, and a mechanism to allow humans to modify the tree in simple ways, including tree deepening, decision variable modification, and leaf node modification. The following conditions: **IV1-C2: AI-Led Policy Modification**, **IV1-C3: Static Policy - Interpretability**, and **IV1-C4: Static Policy - Black-Box** all utilize the same architecture and starting policy but ablate different components of the interaction and interpretability.

After a teaming episode in the **IV1-C2: AI-Led Policy Modification** condition, the AI utilizes recent gameplay to fine-tune a human gameplay model via Behavioral Cloning and performs reinforcement learning for five minutes to optimize its own policy to better support the human teammate. In this collaborative agent policy optimization stage, we utilize the parameters described in Section C.4 and add a timer to stop the optimization. Upon completion of policy optimization, we check if the policy has improved through simulated interactions with the behavior cloning agent, and if so, update the policy. In the case that the policy degrades, we use the original policy prior to optimization. The user can visualize the updated AI policy in its interpretable tree form prior to the next teaming interaction.

**IV1-C3: Static Policy - Interpretability** and **IV1-C4: Static Policy - Black-Box** are static policies that do not change across repeated gameplay. Thus, we do not have any specific additional hyperparameters to discuss within the appendix.

To improve the transparency of the conditions in our experiment, we provide a flow diagram that displays the interaction being assumed within each condition in Figure 9.

### D.3 Compensation Scheme

Participants were compensated at a rate of 20 US dollars per hour of the experiment.

## E Complete Statistical Analysis

Here, we present complete details regarding our analysis, including all test statistics as well as nonsignificant and trending comparisons.

### E.1 RQ1: Team Coordination Performance

As mentioned in the main paper, we allow humans to team with the AI across four episodes, providing us with four teaming scores. Within the main paper, we reported differences with respect to the maximum score participants were able to obtain across iterations. Here, we analyze data in the performance round (the last iteration), where participants were told to maximize performance. We note that participants self-reported their gaming familiarity (100-point scale) and weekly hours playing video games. Across all participants, self-reported gaming familiarity was rated as $73.19 \pm 23.80$ and weekly gaming hours was $4.44 \pm 5.32$. This information was used in our statistical analysis, and significance was not found in performance variation as a function of gaming expertise. Utilizing a Friedman's test, we find that there is a significant difference across domains ($\chi^2(1) = 38.7, p < 0.001$). Accordingly, we analyze the two domains separately.

In **IV2-D1**, we find our data does not meet the necessary assumptions and utilize non-parametric tests. A Kruskal-Wallis Test was conducted to analyze differences in performance round reward across conditions, and we find a significant effect ($\chi^2(4) = 20.85, p < 0.001$) across conditions. We conduct post-hoc pairwise comparisons, utilizing Dunn's test, and find that **IV1-C5** is significantly better than **IV1-C1** ($p < 0.01$), **IV1-C3** ($p < 0.01$), and **IV1-C4** ($p < 0.01$). **IV1-C5** is trending as significantly better than **IV1-C2** with a p-value of 0.0275 (significance is $< 0.025$ or ($\alpha/2$) due to the Bejamini-Hochberg adjustment).

In **IV2-D2**, we test for normality and homoschedascity and do not reject the null hypothesis in either case, using Shapiro-Wilk ($p > .50$) and Levene's Test ($p > 0.05$). An ANOVA was conducted to analyze differences in performance round reward across conditions, taking several observed variables into account. We find a significant effect ($F(4, 38) = 18.93; p < 0.001$) across conditions and decision tree familiarity ($F(1, 38) = 16.12; p < 0.05$). We conduct post-hoc pairwise comparisons, utilizing Tukey HSD, and find that 1) **IV1-C5** is significantly better than **IV1-C2** ($p < 0.01$), **IV1-C3** ($p < 0.01$), and **IV1-C4** ($p < 0.01$), and 2) **IV1-C1** is significantly better than **IV1-C3**.

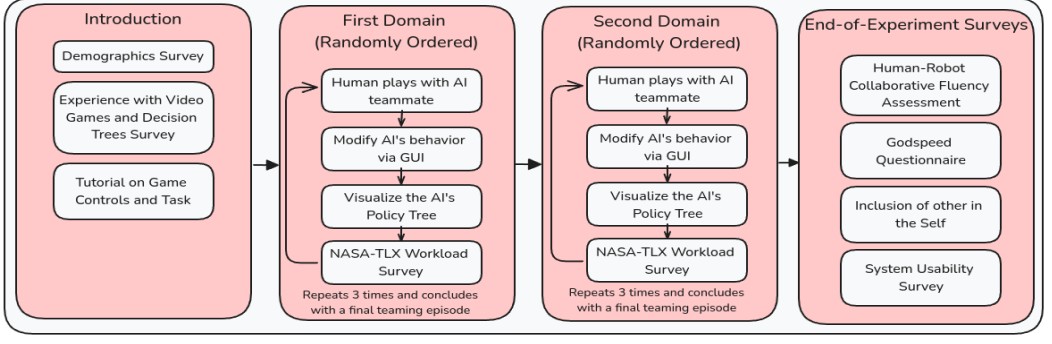

(a) Experiment Flow for IV1-C1: Human-Led Policy Modification

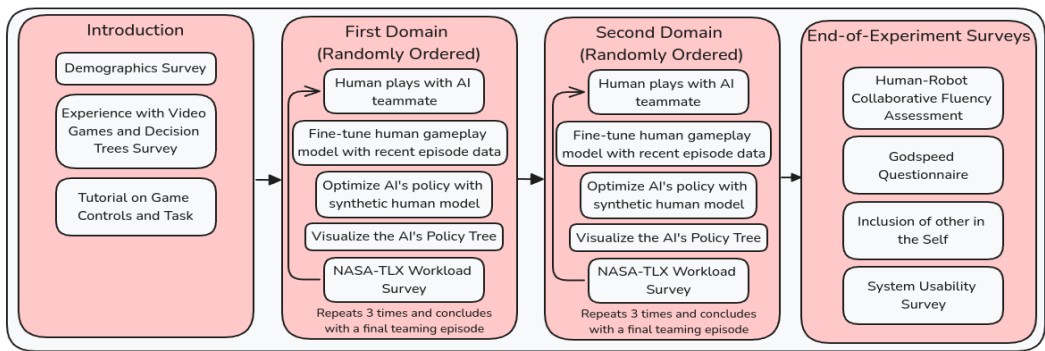

(b) Experiment Flow for IV1-C2: AI-Led Policy Modification

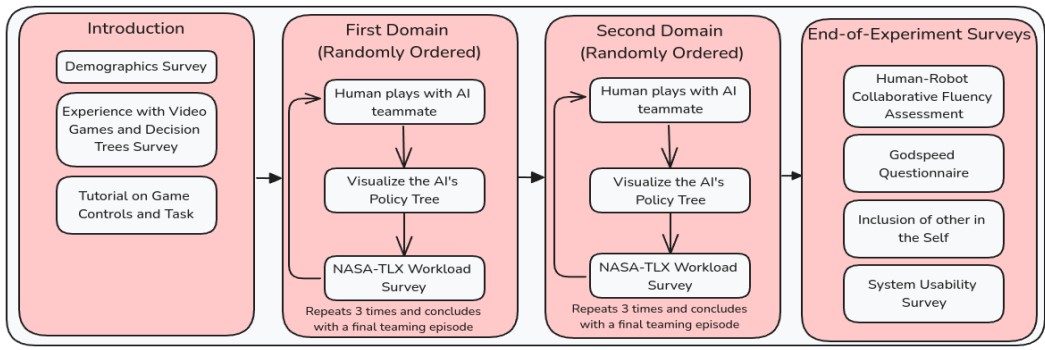

(c) Experiment Flow for IV1-C3: Static Policy - Interpretability

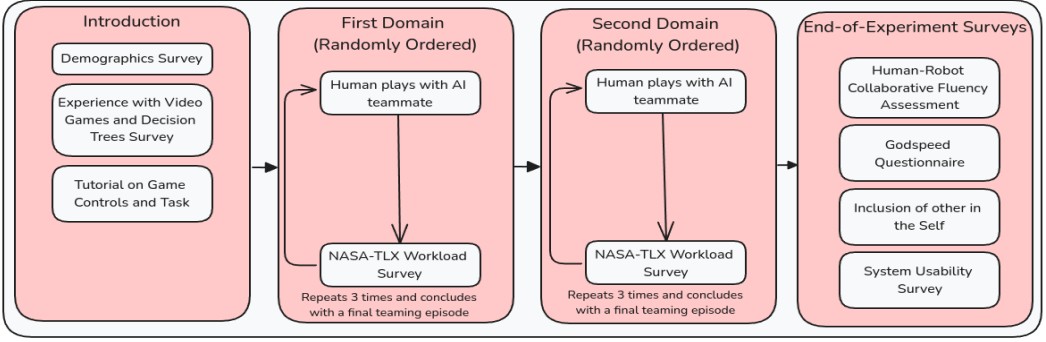

(d) Experiment Flow for IV1-C4: Static Policy - Black-Box and IV1-C5: Fictitious Co-Play

Figure 9: This figure displays an experiment flow diagram for each condition.

These results are similar to those in the paper when analyzing the maximum reward and result in a similar set of conclusions: 1) black-box models can outperform white-box approaches, and 2) white-box approaches with policy modification have some benefit over white-box approaches alone. Further, as we see that tree familiarity positively correlates with performance round rewards, exploring alternative paradigms, such as natural language for describing and programming trees may benefit users unfamiliar with decision trees.

### E.2 Team Development

Here, we analyze the trends across iterations (did agents improve from iteration one to four) and identify characteristics of users that performed well in team development. Utilizing a Friedman's test, we find that there is a significant difference across domains ($\chi^2(1)$=20.48, $p < 0.001$).

We conduct separate Wilcoxin signed-rank tests for each condition, and utilize the Bonferroni correction in determining significance ($\alpha/5$). In **IV2-D1**, we see no condition significantly improves significantly over repeated iterations. In **IV2-D2**, we find that **IV1-C1** ($p < 0.01$) and **IV1-C2** ($p < 0.01$) significantly improve over repeated teaming interactions.

## F  Discussion, Limitations, Future Work, and Societal Impacts

**Discussion:** In this paper, we provide several contributions towards interactive HMT. We first present weaknesses in prior work, displaying that learned collaborative agents can be individualistic and rigid. To address these weaknesses, we propose an interactive scheme termed human-led policy modification to bridge the gap between individualized coordination and adaptive, effective collaboration. We do so by creating a feedback loop that facilitates team policy changes during HMT. This is accomplished by 1) utilizing an interpretable policy representation to provide users with insight into the teammate's decision-making, specifically the IDCT, an interpretable tree-based model that can be trained via reinforcement learning and pruned to a smaller, equivalent representation, and 2) creating a user interface to support the end-user modifying the policy to their evolving specifications. We deploy and compare our interactive policy modification scheme to several other techniques, including two popular prior works and variations of our proposed condition. While we do not a direct objective benefit of human-led policy modification compared to utilizing a black-box model supported with a population-based training scheme [40], we find important takeaways that motivate the importance of conducting longer-term, repeated-interaction studies. Specifically, white-box approaches that facilitate interpretation can be used within a feedback loop to lead to policy improvement, users may require a larger number of interactions to reach a team consensus and maximal performance, and there are person-specific characteristics that may lead to some users being able to take advantage of interpretable models and interaction more than others.

**Limitations:** This study was conducted at a university. While the population was diverse in age, gender, and university major, all students had some college education and most students were based in engineering, presenting a population bias. Furthermore, the population represented by the age group of 18 to 32 years old (mean of 24.14, std of 4.1) within our experiment may not directly generalize to an older population with extensive training. Furthermore, the experiment findings may not generalize to all contexts and scenarios within HMT. We reiterate that our findings are within a two-agent human-machine team within a relatively low-dimensional and short-horizon game, Overcooked-AI. In scaling to more complex and dynamic environments, the tree size needed to represent a high-performing agent will likely increase. In these cases, users may require more time to interact with and understand an agent's policy. There may be several capabilities that can be added to the Human-Led Policy Modification interaction paradigm, which may make the process quicker and easier. For example, model verification or forward simulation can be used to provide the human with other types of feedback prior to the next teaming iteration. Furthermore, for increasingly complex games, agent policies can also operate over different levels of abstraction, providing the human with a tradeoff with fine-grained control of the agent policy and tree size. Finally, different policy visualizations may better support certain populations of users, emphasizing the need for collecting user background information and future research in interpretability for embodied agents.

**Future Work:** In the future, it would be interesting to conduct a similar experiment to a higher number of iterations, or until the team converges to a set of coordination strategies (the "performing" stage in Tuckman's model). Further, the possibility of adding in feedback from the AI regarding human-led

policy modification (checking for logic inconsistencies, etc.) may be used to facilitate speedier team development. It would also be interesting to utilize different paradigms in communicating with the human as language may be an easier medium than a decision tree interface. Future work should also be done to optimize the accessibility of GUIs for policy modification via xAI techniques. Finally, expanding this research to real-world collaborative robot settings in healthcare of manufacturing that utilize tree-based policies, such as collaborative packaging [12, 17] or agile robotics [8, 21], would lead to additional insight into human-machine team development with robot teammates.

**Positive and Negative Societal Impact:** This work investigates repeated interactions with interpretable machine-learning-based agents in a collaborative game. As autonomous agents (e.g., robots) are deployed in the real world, insights from this work may be applied to assist in creating a fruitful working relationship between a human and an agent. We do not believe this work has any negative societal impacts.

# G    Working Definition of Interpretability

As mentioned in the main paper, our agent representation is that of an Interpretable Discrete Control Tree, which reasons over a state space with high-level binary features and multi-step macro-actions. This model (which, in layman's terms, is a decision tree with action probabilities at each node) is the true learned model produced via reinforcement learning, not an abstraction created post hoc. This model is interpretable as its representation is "constrained in model form so that it is either useful to someone, or obeys structural knowledge of the domain, such as monotonicity, causality, structural (generative) constraints, additivity, or physical constraints that come from domain knowledge" [35]. In our case, the model constraints are inherent within the novel IDCT architecture, and the utility of this model to a user is that this model 1) is able to provide users with some awareness over the agent's behavior (and possibly, simulate the agent's decision making) and 2) provides users with the ability to explicitly modify agent behavior (a capability not possible with black-box models).

