# OpenReview forum: "Designs for Enabling Collaboration in Human-Machine Teaming via Interactive and Explainable Systems"
_NeurIPS.cc/2024/Conference — NeurIPS 2024 poster_

### Official Review · Reviewer_MFRs · 2024-06-23

**Soundness:** 4
**Presentation:** 4
**Contribution:** 4
**Rating:** 8
**Confidence:** 3

**Summary:**

The paper proposes a white-box approach to human-machine teaming, in which human teammates can see their virtual counterparts’ policies and adjust them accordingly. The framework is built on top of differentiable decision trees, and the authors propose contextual pruning as a means of simplification for training purposes and readability by the human team members. They use this framework to quantify the benefits of interpretability in HMT and the tradeoff between high interpretability and high accuracy, the latter typically better achieved by traditional black box solutions. The authors use their framework to conduct a thorough statistical analysis comparing different white and black box conditions and use their findings to list a series of guidelines for HMT.

**Strengths:**

- I enjoyed reading this paper, and I think the idea is novel, and the contribution of this paper to the field is significant. Interpretability is crucial in HMT if virtual agents are to be seen as real teammates instead of mere tools. Being able to understand these agents' policies and change them to agree with the human's beliefs is one step in that direction. The key point here is that the changes made by humans may not be the best ones (typically achieved by blackbox solutions), but they may be the ones that foster more collaboration between humans and machines.
- The paper is very well-written, and the ideas are very clear. The steps for reproducibility are detailed, and the statistical analysis is comprehensive.

**Weaknesses:**

The authors propose a pruning strategy to simplify training and interpretability but provide no evidence supporting either. It would be helpful to validate such claims with time and memory usage assessment and IV1-C1 analysis with and without pruning. The authors briefly discuss the former in C.3.2, but it could be improved by displaying the numbers.

**Questions:**

- In line 230, the authors state that contextual pruning significantly improves the ease of training. Are the two pruning strategies enough? I am unfamiliar with DDT, so I wonder if this is novel. How is this different than traditional pruning methods in DDTs?
- As noted by the authors in line 357, some participants outperformed the maximum performance of IVI-C5 in teaming iterations 3 and 4. I wonder if those participants show an increasing trend in performance from iteration 1 to 4? I cannot tell that from the scatter plot. I am asking this because I am curious to know if the improvements are due to randomness or if humans are applying a conscious strategy when modifying the agents’ policies, leading to incremental improvements at every iteration.

**Limitations:**

The authors discuss the limitations of their work in the Appendix. I would argue there's one more limitation:

It seems performance in the mixed-initiative scenario is highly dependent on presentation, as validated by better performance improvements in participants more familiar with Trees. This work shows that interpretability is also relative. What is interpretable to me may be convoluted to another, which may also affect the team's performance. It would be interesting to check for differences in IVI-C1 with different visualization methods in future work.

---

> ### Author Rebuttal · Authors · 2024-08-06
>
> We thank the reviewer for noting that they enjoyed reading our paper, that the paper's contribution to the field is significant, and that our paper is well-written and the ideas are clear. We have responded below to the weaknesses and questions noted by the reviewer.
>
> **Contextual Pruning Results and Novelty** -- The two post-hoc pruning strategies we utilize are to prune a tree given the boundaries of each state variable and removing redundancies within the tree. This need for pruning arises naturally during the training of the IDCT model via reinforcement learning as different subspaces of the tree model may become more important (higher probability of reaching certain sub-trees) or completely inactivated (impossible to reach a certain sub-space due to changes in weights caused by gradient descent). It is important to note that these pruning techniques follow a similar ideology behind neural network pruning but do not modify the model's behavior in any way. Neural network pruning approaches, on the other hand, often remove weights with smaller magnitudes or activations, which result in model changes that can harm performance.
>
> These specific pruning strategies have not been applied to differentiable decision tree models before to improve interpretability. In training the IDCT model, we conducted a hyperparameter search over different tree sizes and chose the best-performing model post-pruning with under 32 leaves. In the end, we found that a 256-leaf IDCT model was most amenable to training in each domain, and contextual pruning was able to reduce this large tree into a simple tree of three leaves and two leaves, respectively (a 64-128x reduction in model size). This pruning technique is essential as trees of arbitrarily large depths can be difficult to understand [1] and simulate [2], and that a sufficiently sparse DT is desirable and considered interpretable [3].
>
> [1] Abhishek Ghose and Balaraman Ravindran. Interpretability with accurate small models. Frontiers452
> in Artificial Intelligence, 3, 2020.
>
> [2] Himabindu Lakkaraju, Stephen H. Bach, and Jure Leskovec. Interpretable decision sets: A454
> joint framework for description and prediction. In Proceedings of the 22nd ACM SIGKDD455
> International Conference on Knowledge Discovery and Data Mining, KDD ’16, page 1675–1684,456
> New York, NY, USA, 2016. Association for Computing Machinery.
>
> [3] Zachary C Lipton. The mythos of model interpretability: In machine learning, the concept of458
> interpretability is both important and slippery. Queue, 16(3):31–57, 2018.
>
> **Human Improvement from Iteration 1 to 4** -- In the rebuttal document, we have added in an additional display of the reward trajectories for each participant, with a single color assigned to each participant's behavior for each domain. These figures, while showcasing the trajectories of each participant well, can get
> cluttered and thus we chose to showcase our findings via a scatter plot. In IV2-D1: Forced Coordination, you can see that many participants do not linearly increase from Iterations 1 to 4. Often, mistakes in tree logic can lead to an agent that does not collaborate well and as this domain requires collaboration, leads to a sub-100 score.
>
> The improvement in IV2-D2: Optional Collaboration between iterations one and four was found to be significant (p<0.01). This implies that that in this domain, users were applying a conscious strategy when modifying the agents policies. However, even in this domain, the improvement was not necessarily monotonically increasing for all participants.
>
> **Additional Limitation** -- We thank the reviewer for noting this additional limitation and will add this into our section.

---

> > ### Comment · Reviewer_MFRs · 2024-08-08
> > **I maintain my decision**
> >
> > Thank you for the rebuttal and additional results. I think this is a good paper. I maintain my decision.

---

### Official Review · Reviewer_cdZ4 · 2024-07-11

**Soundness:** 2
**Presentation:** 2
**Contribution:** 2
**Rating:** 6
**Confidence:** 4

**Summary:**

In this paper the authors present an approach to human-AI teaming in the common Overcooked domain via Interpretable Discrete Control Trees (IDCTs), which are differentiable decision trees which the authors visualize and make controllable. The authors present two examples of where existing blackbox models may demonstrate a gap in teaming performance. They then present results of a human subject study that demonstrates that authors' approach is outperformed by a fictitious co-play baseline.

**Strengths:**

Human subject studies studying human-AI teaming are still relatively rare due to their difficulty. As such, a new study is always beneficial and of interest to the community. The authors also include many variations of their approach, which is beneficial in terms of more deeply understanding the mechanisms of human-AI teaming.

**Weaknesses:**

I like this work, but at present this paper has a number of key weaknesses holding it back.

First, the current text of the paper contains a large number of unsupported claims. Almost all of the text in italics represents claims that are not substantiated by an argument, a citation, or the results of the paper. There are similarly design decisions that are not motivated or explained, such as why the authors chose to use IDCTs.

Second, Section 3 introduces the possibility of a gap in teaming performance with current models. However, the evidence for this is presented as two examples. This is not sufficient to demonstrate that this specific approach has failure cases but not to demonstrate a need for the approach the authors' propose. I'd recommend working on a more full survey of existing Overcooked approaches to quantitatively evaluate the rate at which collaboration gaps may be occurring. This section represents the primary motivation for the authors' work, and so it being a weak point makes the whole paper weaker.

Third, the authors do not give full technical details for any of the approaches, not the approach of Carroll et al., nor their own ICDT approach, nor their implementation of fictitious co-play. This is an issue as it makes it very difficult as a reader to understand what has been done at a technical level. For example, it's unclear to me what size the ICDTs used in the study were, I only know that the users were limited to expanding the tree to a depth of 4. But this doesn't tell me if the initial tree had a depth of 3. Similarly, the pruning approach only seems to remove redundant nodes so it's unclear how much this would actually prune (the authors state 8-16x smaller but it's unclear to which trees they refer). Given that a major problem with the ICDT's performance in the user study was their performance, these details are crucial. While there's an anonymized GitHub in the appendices, it should not be necessary to go through it to understand the work.

Fourth, while the human subject study seems well-designed, certain details are unclear. It's unclear what population is being drawn on for the participants, though I would guess university undergraduates given the demographics. It's unclear how knowledgeable these participants were about AI except for one mentioned figure around familiarity with trees. It's unclear why the authors stopped recruiting at 50, since 10 per condition is considered an absolute minimum. Finally, the methodology is somewhat unclear, its unclear what order the participants experienced the domains. Was it consistent, if so, why? If not, there were 10 conditions, not 5.

Finally, the results do not seem to support the authors' claims. From the result, my takeaway is that participants preferred performance over interpretability. This goes against several of the authors' claims in the paper, most notably "As seen through these findings, the ability to interact with an interpretable model is perceived significantly better across several measures". The authors also make a claim about trending towards significance, which is not statistically sound.

**Questions:**

1. What was the size of the ICDTs used in the study?
2. Did participants experience the two domains in a fixed or random order?

**Limitations:**

The major limitations come from the poor performance of the ICDTs and the decision to limit the size of the human subject study. The authors do not sufficiently address these limitations.

---

> ### Author Rebuttal · Authors · 2024-08-06
>
> We thank the reviewer for noting that our study in Human-AI teaming would be beneficial to the community.
>
> **Unsupported Claims and Design Decisions** -- As the field of Human-AI Teaming is still relatively new, some motivations are positional. For example, in considering lines 129-132, some may argue that AI agents and humans do not need consensus or that the human should always defer to the AI's strategy. In some domains, this may be effective. However, in domains where agents and humans need to collaborate closely, we believe the process of consensus will lead to better collaboration performance and understanding between teammates. This is validated in our human-subject study findings (see Figure 4b of the manuscript), where users that can interact with the AI's policy significantly improve over repeated gameplay. There are several works that display that social robots that can adapt online and develop social relationships achieve more successful and sustained interactions with users [1].
>
> [1] Leite et al. Empathic robots for long-term interaction: evaluating social presence, engagement and perceived support in children.
>
> The design guidelines are directly related to the results of our paper.
>
> - Creation of white-box agents that achieve competitive initial performance to black-box agents: If model performance is equal, interpretable models provide increased accountability compared to black-box models. Importantly, these models with user interaction led to positive team development.
> - Design of learning schemes to support the generation of collaborative behaviors: This is derived from the case study and analysis of Fictitious Co-Play. Designing objective functions so that agents collaborate well with humans and/or online adaptation schemes to lead to personalized, effective teammates is an important research direction.
> - Creation of interfaces that enable mixed-ability users to improve team collaboration: This is based on system usability scores collected from users, emphasizing the need for creating modification interfaces that support a wider variety of users.
> - Evaluation of teaming in a larger number of interactions: This was derived from our user study findings, where a higher # of interactions may have provided a better understanding of the team development process.
>
> The design decision behind the IDCT is that this tree-based model affords transparency and supports training via RL. This allows us to directly compare with prior frameworks in Overcooked that leverage RL for training collaborative agents and provides us with an interpretable model that users can modify and visualize.
>
> **Case Study regarding the Teaming Gap with Current Models** -- We agree that this case study could be expanded. This first study (Figure 1) is conducted with an actual human player following a scripted strategy while collaborating with an agent publicly available from Carroll et al. While a human could have conducted a larger number of trials with the agent, it was clear from the set of trials that the agent could not adapt to the human-preferred strategy.
>
> The second study focuses on Ficticious Co-Play. A heuristic collaborative strategy and heuristic individual strategy are programmed, receiving scores of 408 and 306. Then, an FCP agent is trained, converging to a score of 295.06 \pm 1.86 over 50 teaming simulations. This FCP agent is also evaluated with humans in the study, achieving scores ranging from approximately 120 to 315 in the last teaming round (Figure 4b-right). These gameplay scores of FCP, both in teaming with synthetic agents and real humans, are far below a heuristic collaborative strategy, signifying the gap. A full assessment including other benchmarks and analyzing agent collaborativeness, both quantitatively and qualitatively, while interesting, would likely be a full paper in itself.
>
> **Technical Details regarding the Approaches** -- The details regarding the IDCT approach are found in Appendix C alongside pictures of the agent models (Figure 6 and 7). In line 321, we note that the trained policies had two leaves in Forced Coordination and three leaves in Optional Collaboration. As mentioned in Appendix C.4, we train our IDCT models with 256 leaves. A reduction to a depth of one (2 leaves) or two (3 leaves) is a pruning reduction of 128x and 64x. We will update these numbers and shift these details into the main paper.
>
> Details of the FCP baseline are also found in Appendix Section C.4. The AI-Led Policy Modification is an adaptation of Carroll et al.'s approach to an online setting with an IDCT. We provided high-level details about this approach in Line 279-284 and within the footnote on page 6. We will include further details about the online optimization procedure in the Appendix.
>
> **User Study Detail Clarifications --** Our study was conducted at a university with a diverse population majoring in different engineering disciplines, economics, and sciences. All users had some college education and were enrolled at an engineering-focused university as an undergraduate or graduate student. Users were asked demographics information, experience with games and decision trees, and conducted a personality survey. This information was included when determining significant trends.
>
> In future studies, it would be beneficial to increase the # of participants and gameplay trials. As noted in the Procedure, the domains are randomly ordered. Our experiment is a 5 (teaming method; between) x 2 (no. of domains; within) x 4 (no. of repeated evaluations; within) mixed-factorial experiment.
>
> **Results Supporting Author Claims** - The statement noted was comparing conditions IV1-C1 to IV1-C2, IV1-C3, and IV1-C4. This statement meant that given the same tree-based model, the ability to interact adds subjective benefits. The reviewer is correct in noting that participants did assess higher-performing agents more positively in their subjective ratings. We will clarify this statement.

---

> > ### Comment · Reviewer_cdZ4 · 2024-08-08
> > **Re: Rebuttal by Authors**
> >
> > Thanks to the authors for their detailed response. My concerns around the claims and the results not supporting them, remains unchanged and so I am choosing to maintain my score.

---

> > > ### Author Response · Authors · 2024-08-08
> > >
> > > Thank you for reading our rebuttal and replying so quickly. We would love to improve our paper by adjusting the language around unsupported claims or better tying certain results to these claims. Above, in our reply, we discussed how our results directly relate to our guidelines. Are there any other specific unsupported claims you would be able to highlight so that we could improve our paper?
> > >
> > > This would be much appreciated. Thank you in advance!

---

> > > > ### Comment · Reviewer_cdZ4 · 2024-08-11
> > > > **Re: Official Comment by Authors**
> > > >
> > > > I apologize for the delay in my response. My primary concern, per my initial review is "As seen through these findings, the ability to interact with an interpretable model is perceived significantly better across several measures". I recognize the comparisons being made, but my concern is that this overstates the outcomes, which appear to be more mixed. I recognize the authors have said they plan to revise the claims but there was no clarification as to how. Clarifying this point would be beneficial for my evaluation.

---

> > > > > ### Author Response · Authors · 2024-08-12
> > > > >
> > > > > Thank you very much for getting back to us and allowing us to clarify this disparity.  In clarifying this specific statement, we propose the following to better ground this claim.
> > > > >
> > > > > "Overall, participants generally assessed higher-performing agents more positively in their subjective ratings. In considering conditions that utilized a tree-based model (IV1-C1, IV1-C2, IV1-C3, and IV1-C4), we see the addition of interaction with the tree policy provides significant subjective benefits in positive teaming traits and collaborative fluency (defined within the Human-Robot Collaborative Fluency Assessment [12]). In including the remaining condition which utilizes a black-box model, IV1-C5: Fictitious Co-Play, and comparing it to IV1-C1: Human-Led Policy Modification, we see that even though Fictitious Co-Play outperformed Human-Led Policy Modification in terms of team reward (though not significantly in the domain of Optional Collaboration), no significant subjective differences were observed between these two conditions. This presents an interesting relationship between transparency, interaction, and performance in its relation to subjective perception that warrants future research."
> > > > >
> > > > >
> > > > >
> > > > >
> > > > > We will also update the first design guideline to the following (the change is bolded). This change should also contribute to reducing the previous overstatement.
> > > > >
> > > > > 1. *The creation of white-box learning approaches that can produce interpretable collaborative agents that achieve competitive initial performance to that of black-box agents.* This guideline is critical to providing humans with the subjective benefits **obtained from interactivity with** white-box models, objective benefits of black-box models, and the ability to interact with policies to facilitate team development.
> > > > >
> > > > >
> > > > > Again, thank you for helping us improve our manuscript.

---

> > > > > > ### Comment · Reviewer_cdZ4 · 2024-08-12
> > > > > > **Re: improved claims**
> > > > > >
> > > > > > Thanks for taking the time to clarify those points! I think that is much improved from the version in the paper. Given this and the other information presented in the rebuttal and this discussion, I have increased my score to a 6. I do not expect it will increase further as the remaining concerns are difficult to impossible to address in the rebuttal period, namely the relatively small participation pool/reference points for both the teaming gap studies and final human subject study.

---

### Official Review · Reviewer_HJn7 · 2024-07-11

**Soundness:** 3
**Presentation:** 3
**Contribution:** 3
**Rating:** 5
**Confidence:** 4

**Summary:**

In this manuscript, it focuses on the collaboration in human-machine teaming (HMT) based on interactive and explainable systems. In order to address the existing issues, such as decoupling, the author(s) explored an interesting paradigm in HMT and proposed some guidelines based on the study. Also, the author(s) pointed out some potential research directions, although some of the conclusions are intuitive, but interesting.

**Strengths:**

In my opinion, the strengths of this manuscript are as follows:

1. Demonstrating that current HMT approaches struggle to adapt to human-preferred strategies, often resulting in suboptimal team performance.

2. The proposed architecture enables end-users to modify AI behavior, facilitating a feedback loop for team development.

**Weaknesses:**

In my opinion, the weaknesses of this manuscript are as follows:

1. As the author(s) mentioned, due to the population setting, the findings have a population bias and may not generalize to wider situations.

2. The cited work is a little out of date. This makes this manuscript seem not ready for publishing at conferences such as NeurIPS.

3. Some descriptions in the main text are not so clear.

For more details, please see the Section "Questions" below.

**Questions:**

I read the manuscript, and I have the following questions/comments. Thanks.

1. We need to balance sparsity and stability in designing a learning algorithm. In this manuscript, the author(s) adopted L1 regularization; I am not so sure if it will cause some issues due to the instability of l1.

2. Regarding how users interact with AI across repeated play under different factors, the author(s) performed some human-subject-involved studies. About the population used in the study, can you discuss the implications of the population bias in more detail and how it might affect the generalizability of your findings?
Also, maybe sex/age-based difference analysis is also worth exploring.

3. If possible, could you explain more about the potential challenges in scaling the human-led policy modification approach to more complex and dynamic environments beyond the Overcooked-AI domain?

4. Regarding the citations, I do not think the current references are sufficient, at least, for a manuscript preparing to submit to computer conferences such as NeurIPS, since the newest citation was two years ago, and there are only 4 such 2022 citations.

5. In Line 58, maybe "reinforcement learning" should be with its abbreviation, not until Line 70.
6. It would be great to be with the full name for PPO, such as Proximal ... (PPO)

Some other format issues in references:

(1) The author's name, sometimes using the abbreviation, sometimes not, such as Ref.[1], Ref.[33].

(2) Sometimes, the conference name has both the full name and its abbreviation, and sometimes not, such as, Ref.[4] vs Ref.[5].

(3) In Ref.[17], "ai"=>"AI".
Please check carefully; it would be great if the author(s) could correct these issues.


I would like to consider adjusting my current score based on the responses from the author(s). Thanks.

**Limitations:**

Yes.

---

> ### Author Rebuttal · Authors · 2024-08-06
>
> We thank the reviewer for their valuable feedback. We have responded to the weaknesses and questions noted by the reviewer.
>
> **Instability of L1** -- We thank the reviewer for this comment. The L1 regularization is only applied to the action leaf nodes of the tree policy. This regularization serves to make the categorical distributions within each leaf node more sparse so that the agent acts more deterministically. The mechanism used in our architecture to train an interpretable tree model via reinforcement learning is called differentiable crispification and was formulated by prior work (see [1]). From our observation, the L1 regularization successfully improved how deterministic the resultant tree-based policies were.
>
> [1] Paleja, R., Niu, Y., Silva, A., Ritchie, C., Choi, S., & Gombolay, M. (2022). Learning interpretable, high-performing policies for continuous control problems. arXiv preprint arXiv:2202.02352.
>
> **Population bias and how it might affect the generalizability of your findings? Also, maybe sex/age-based difference analysis is also worth exploring.** -- Our study was conducted at a university with a diverse population majoring in a variety of different engineering disciplines (Electrical and Computer, Biomedical, Aerospace, Chemical and Biomolecular), economics, and variety of sciences (computer science, neuroscience, atmospheric). All users within our experiment had some college education and were enrolled at an engineering-focused university. As such, these findings may not generalize beyond this population. In our data collection, we collected demographic information including age and sex. In our analysis, these variables were used and we did not find any significant trends that displayed that age or sex led to differences in Human-AI collaboration performance.
>
> In generalizing our results to a broader population, we believe are findings can be augmented by other literature in human-AI interaction to design better interfaces for specific populations or domain-specific collaborative agents.
>
> **If possible, could you explain more about the potential challenges in scaling the human-led policy modification approach to more complex and dynamic environments beyond the Overcooked-AI domain?** -- Thank you for this excellent question. In scaling to more complex and dynamic environments, the tree size needed to represent a high-performing agent will likely increase. In these cases, users may require more time to interact with and understand an agent's policy. There are also several capabilities that can be added to human-led policy modification which may make the process quicker and easier. For example, model verification can be added into the tree modification interface to detect problems with logic (detect regions of the tree that cannot be reached). This would allow the human to receive other types of feedback prior to teaming with the AI. For complex games, agent policies can also operate over different levels of abstraction providing the human with a tradeoff with fine-grained control of the agent policy and tree size. There may also be other more accessible mediums, such as language, that the human can use to program a large tree policy.
>
> **Missing citations** -- We thank the reviewer for noting this deficiency. Within our literature review, we will add the following recent papers that are closely related to our work.
>
> [2] Hong, Joey, Sergey Levine, and Anca Dragan. "Learning to influence human behavior with offline reinforcement learning." Advances in Neural Information Processing Systems 36 (2024).
>
> [3] Wang, Chenxu, et al. "On the Utility of External Agent Intention Predictor for Human-AI Coordination." arXiv preprint arXiv:2405.02229 (2024).
>
> [4] Guan, Cong, et al. "One by One, Continual Coordinating with Humans via Hyper-Teammate Identification."
>
> [5] Tulli, Silvia, Stylianos Loukas Vasileiou, and Sarath Sreedharan. "Human-Modeling in Sequential Decision-Making: An Analysis through the Lens of Human-Aware AI." arXiv preprint arXiv:2405.07773 (2024).
>
> **Abbreviations and Reference Format ** -- We have updated our paper language per your comments and updated our references to be consistent.
>
>
> **Ethics Review Flag** -- As mentioned in Line 342 of our paper, our experiment was approved by a university institutional review board (IRB). All participants in our experiment signed a consent form, received description of the risks involved in our study, and received compensation.

---

> > ### Comment · Reviewer_HJn7 · 2024-08-10
> > **Updating**
> >
> > Thanks for the responses from the author(s). The responses from the author(s) clarified some of my concerns to some extent. I increased my score from 4 to 5.

---

### Official Review · Reviewer_yVVF · 2024-07-12

**Soundness:** 2
**Presentation:** 2
**Contribution:** 2
**Rating:** 5
**Confidence:** 3

**Summary:**

This paper focuses on developing strategies to enhance transparency and interpretability in human-AI teaming settings. Based on my understanding, two collaboration contexts have been considered: human-preferred collaboration and AI-preferred suboptimal teaming strategy. The authors have implemented specific strategies to tackle these contexts and address interpretability. The implemented strategies have been evaluated on the Overcooked game with real human users.

**Strengths:**

The paper is well-motivated, particularly in its approach to learning human preferences and using reinforcement learning (RL) to train the agent.

The paper provides valuable insights into designing human-AI collaboration interfaces to enhance user trust and acceptance, addressing a timely and relatively unexplored domain.

**Weaknesses:**

The main weakness of the paper is that it is very hard to follow, and the methodology lacks transparency. For instance, the paper does not present a single method to investigate explainability. Overall, the chosen modifications and the designed collaboration setups are not clearly presented. It is not straightforward to link Section 4 (Methodology) with Section 5 (Studies).

Similarly, the experimental results are very hard to follow. The numbers are presented but are not straightforward to interpret or use for cross-checking the claims. In particular, the evaluation metrics are not clear.

Additionally, only one collaboration environment has been considered. The findings might be biased toward this specific collaboration task.

**Questions:**

Three main questions are:

- How do you envision improving the transparency of the proposed methodologies? How can Section 4 be better linked with Section 5?
- What are the evaluation metrics for the chosen collaboration settings? How is interpretability evaluated?
- Are there any other collaboration environments that can be considered for this line of research? How do you envision these findings being generalized to other collaboration tasks?

**Limitations:**

The authors discussed the limitations and the broader impact of their work adequately in the appendix.

---

> ### Author Rebuttal · Authors · 2024-08-06
>
> We thank the reviewer for noting that our paper is well-motivated, provides valuable insight into designing human-AI collaboration interfaces, and studies a relatively underexplored domain. We have responded to the weaknesses and questions noted by the reviewer.
>
> **Comparison to Explainability Approaches** --  The reviewer is correct in that we do not compare against explainability approaches that utilize local explanations to explain the decision-making of autonomous agents. While these approaches should be tested to explain behavior in the tight human-AI collaboration settings we consider, local explanations can be misleading and may not accurately represent the agent's decision-making behavior. This would lead to another dimension that needs validation. As this area of research (transparency and adaptability of collaborative agents in repeated human-AI collaboration) is relatively new, we hope our research can spawn further studies of this kind.
>
> **How can Section 4 be better linked with Section 5?** -- Section 4 presents an interpretable machine learning architecture to train collaborative AI teammates, Interpretable Discrete Control Trees (4.1), a training advancement to enhance interpretability, and a mechanism to allow humans to modify the tree in simple ways, including tree deepening, decision variable modification, and leaf node modification.  The creation of this capability (interpretable tree architecture + human modification) is our main proposed condition, IV1-C1: Human-Led Policy Modification in Section 5.
>
> The following conditions IV1-C2: AI-Led Policy Modification, IV1-C3: Static Policy - Interpretability, IV1-C4: Static Policy - Black-Box all utilize the architecture introduced in Section 4 but ablate the interaction and interpretability. We tried to depict this gradual feature reduction across our proposed conditions in Table 1. To improve the linkage between Section 4 and Section 5, we propose to create a new section between 4 and 5 that presents information regarding the training and results of the IDCT model. In creating this section to improve clarity, we would augment details from the author response alongside the paper lines 301-321.
>
> **How do you envision improving the transparency of the proposed methodologies?  --** To improve the transparency of the proposed methodologies, we have created a flow diagram (uploaded as Figure 1 in the attached rebuttal pdf) of each condition that helps display visually the experiment flow and interaction being assumed within the proposed approach. We have also included a diagram of how IDCT agents are generated as part of Figure 2 (left). Both of these figures will be added into the main paper as part of the appendix and alongside the new aforementioned section will improve the paper's clarity.
>
> **What are the evaluation metrics for the chosen collaboration settings?** -- The main objective evaluation metric used to evaluate the performance of the collaboration is the game score. Section B of the appendix describes the exact scoring function that makes up the game score. In short, high score bonuses are obtained for full dishes served, and minor score bonuses are given for smaller objectives like filling a pot or picking up a dish. In the first domain, Forced Coordination, successful dish serving is not possible without resources being handed between the AI and human. In the second domain: Optional Collaboration, agents can serve dishes without explicitly collaborating with the human, but this domain was intentionally designed such that collaboration with the human via timely resource handoffs would result in a higher score than without collaboration.
>
> **How is interpretability evaluated?** -- The interpretability of the model isn't evaluated explicitly. In our hyperparameter search for training the tree-based agent policy, we choose the highest-performing model with under 32 leaves. Prior work [1] notes that there is a cognitive limit on how complex a model can be while still being human-understandable and thus, we prioritized selecting a high-performing IDCT model that was still relatively small. We note that as conditions IV:C1-C4 utilize the same model, utilizing interpretability metrics such as tree size do not provide additional information in understanding the tradeoff between these conditions.
>
> [1] Lakkaraju et al. Interpretable decision sets: A joint framework for description and prediction.
>
> **Are there any other collaboration environments that can be considered for this line of research? How do you envision these findings being generalized to other collaboration tasks?** -- We believe research in repeated teaming with collaborative, transparent agents should be studied in more complex domains such as Minecraft, Dota 2, and Starcraft.  Our work utilizes the relatively low-dimensional Overcooked setting where techniques like online optimization and tree manipulation by a human end-user can be done within a short time period. This is vital in allowing for a feasible single-session user study to be conducted.
>
> However, many of our takeaways generalize beyond our single setting of Overcooked to the field of human-AI collaboration and also have applicability to real-world applications in collaborative robotics. In many settings, teams of agents and humans will need to go through stages of team development. Our work shows that model transparency and the ability to interact with policies help in this regard. While there may be domains where white-box models are already competitive with the initial performance of black-box models, there is still research needed to improve interfaces so that users can interact successfully with agent models. Within our domain, we found through a usability survey that there was a large disparity between users finding the interface good (>75) and bad (<35). This finding also generalizes to many domains that have human collaborators coming from different backgrounds and expertise levels.

---

> ### Comment · Reviewer_yVVF · 2024-08-11
> **reply: Rebuttal by Authors**
>
> After considering all the feedback and responses, while I am not fully convinced of the paper's methodological strength, the idea and its execution are both original and solid. The paper offers insightful findings, which motivated me to raise my score.

---

### Author Rebuttal · Authors · 2024-08-06

We would like to thank all reviewers for their insightful reviews and valuable feedback on our paper. We have included a rebuttal document with additional figures as well as provided rebuttals to each reviewer below.

---

### Decision · Program_Chairs · 2024-09-25

**Decision:**

Accept (poster)

**Comment:**

All the reviews for this paper are positive, even though 2 of them are borderline. These reviewers raised issues that have been somewhat addressed in the author's rebuttal. However, there are still issues with the clarity and generality of the results.

Nonetheless, all reviewers agree that the paper is quite interesting and covers a very relevant topic. I recommend acceptance as a poster.

Finally, I recommend that the authors thread lightly regarding unsupported claims, as discussed with Reviewer cdZ4.